# Impact of Growth Conditions on High-Throughput Identification of Repurposing Drugs for *Pseudomonas aeruginosa* Cystic Fibrosis Lung Infections

**DOI:** 10.3390/antibiotics13070642

**Published:** 2024-07-12

**Authors:** Giovanni Di Bonaventura, Veronica Lupetti, Arianna Pompilio

**Affiliations:** 1Department of Medical, Oral and Biotechnological Sciences, “G. d’Annunzio” University of Chieti-Pescara, 66100 Chieti, Italy; veronica.lupetti@studenti.unich.it (V.L.); arianna.pompilio@unich.it (A.P.); 2Center for Advanced Studies and Technology, “G. d’Annunzio” University of Chieti-Pescara, 66100 Chieti, Italy

**Keywords:** high-throughput screening, drug repurposing, cystic fibrosis, *Pseudomonas aeruginosa*, antibacterial activity, growth conditions

## Abstract

*Pseudomonas aeruginosa* lung infections in cystic fibrosis (CF) patients represent a therapeutic challenge due to antibiotic resistance. Repurposing existing drugs is a promising approach for identifying new antimicrobials. A crucial factor in successful drug repurposing is using assay conditions that mirror the site of infection. Here, the impact of growth conditions on the anti-*P. aeruginosa* activity of a library of 3386 compounds was evaluated. To this, after 24 h exposure, the survival rate of CF *P. aeruginosa* RP73 planktonic cells was assessed spectrophotometrically under “CF-like” (artificial CF sputum, pH 6.8, 5% CO_2_) and enriched (Tryptone Soya Broth, pH 7.2, and aerobiosis) conditions. Among non-antibiotic compounds (n = 3127), 13.4% were active regardless of growth conditions, although only 3.2% had comparable activity; 4% and 6.2% were more active under CF-like or enriched conditions, respectively. Interestingly, 22.1% and 26.6% were active exclusively under CF-like and enriched conditions, respectively. Notably, 7 and 12 hits caused 100% killing under CF-like and enriched conditions, respectively. Among antibiotics (n = 234), 42.3% were active under both conditions, although only 18.4% showed comparable activity; 9.4% and 14.5% were more active under CF-like and enriched conditions, respectively. Interestingly, 23% and 16.6% were active exclusively under CF-like and enriched conditions, respectively. Sulphonamides showed higher activity under CF-like conditions, whereas tetracyclines, fluoroquinolones, and macrolides were more effective under enriched settings. Our findings indicated that growth conditions significantly affect the anti-*P. aeruginosa* activity of antibiotics and non-antibiotic drugs. Consequently, repurposing studies and susceptibility tests should be performed under physicochemical conditions that the pathogen tackles at the site of infection.

## 1. Introduction

In patients with cystic fibrosis (CF), a mutation in the CF transmembrane conductance regulator gene accumulates dry and sticky airway secretions, creating an ideal environment for the onset of pulmonary infections [1]. Among the most common CF pathogens, *Staphylococcus aureus* is common in early life, while *Pseudomonas aeruginosa* is prevalent in adults, causing severe bronchial infections [2].

The changes in the CF lung’s microenvironment—e.g., hypoxia, dysregulated mucus, and neutrophilic inflammation—hinder the clearing of the infection, leading to damage of the airway layer and increasing bronchiectasis, further exacerbating the impaired mucus clearance. This cycle of airway obstruction with recurrent infections and inflammation can eventually lead to the death of patients [1,3].

Chronic lung infection by *P. aeruginosa* is the primary cause of morbidity and mortality in adult CF patients [2]. Repeated cycles of nebulized antibiotics such as tobramycin, colistimethate, levofloxacin, and aztreonam, aimed at reducing the bacterial load and slowing disease progression, can lead to the selection of resistant isolates, making treatment challenging [1,3]. Additionally, the formation of intrinsically antibiotic-resistant biofilms during chronic infection makes exacerbation treatment even more challenging [4]. Therefore, there is an urgent need for new drugs with potent and long-lasting anti-*P. aeruginosa* activity.

Drug development, a complex and costly process [5], is particularly daunting in the realm of rare diseases like CF, where patient numbers are low. However, repurposing existing or investigational drugs offers hope for these patient populations, potentially reducing timelines and costs compared to de novo drug development [6]. Previous studies have shown the effectiveness of this approach in uncovering the antimicrobial potential of drugs with other therapeutic indications—such as antidepressants, antineoplastics, antacids, and hypoglycemic agents—even against multidrug-resistant (MDR) strains [7].

A key factor in successful drug repurposing—particularly in the quest to identify new, clinically relevant antimicrobial compounds—is using assay conditions that mirror the site of infection. Recent studies have highlighted the impact of physicochemical conditions observed in the infected CF lung—such as reduced O_2_ tension, highly viscous sputum, and acidic pH—on the efficacy of various antibiotics [8,9,10,11]. Furthermore, the physicochemical features of the surrounding CF lung environment can drive bacterial physiology. For example, high glucose concentrations in a nutrient-depleted environment, such as those found in CF-related diabetes, can promote the growth of *S. aureus* and *P. aeruginosa* [12]. In addition, factors such as the composition of mucus or the community of other lung microbes can have a major impact on the *P. aeruginosa* phenotype, including antibiotic resistance and virulence traits [13,14].

These observations highlight the potential limitations of routine antibiotic susceptibility testing, typically performed under conditions relevant to the site of infection—i.e., in a rich medium with a slightly alkaline pH and under an aerobic atmosphere—in designing effective antibiotic regimens for *P. aeruginosa*-infected CF patients [15,16].

In the present work, we evaluated the impact of growth conditions on the activity against *P. aeruginosa* by compounds with different therapeutic indications. For the first time, a compound library underwent high-throughput screening (HTS) to assess anti-*P. aeruginosa* activity comparatively under “CF-like” [i.e., 5% CO_2_ atmosphere, artificial sputum medium (ASM), and pH 6.8] and enriched (i.e., aerobiosis, Tryptone Soya Broth, and pH 7.2) conditions.

## 2. Results

### 2.1. Effect of Growth Conditions on the Anti-P. aeruginosa Activity of the Compound Library

All 3386 compounds were tested at a concentration of 0.1 mM under CF-like and enriched experimental conditions. Compounds exhibiting activity were retested in a secondary screening to confirm the potential hits.

The Pearson correlation coefficient indicated a significant relationship between findings obtained under CF-like and enriched conditions (r = 0.150, *p* < 0.0001) (Figure 1). Five hundred twenty (15.3%) compounds exhibited antibacterial activity under both growth conditions tested but to different extents. Specifically, 144 (4.2%) hits showed comparable activity under both conditions (Appendix A), while a significantly lower number of hits was more active under CF-like (147, 4.3%) (Appendix A) conditions than enriched conditions (229, 6.7%; *p* = 0.0001 vs. CF-like) (Appendix A).

A comparable proportion of hits was active exclusively under a growth condition: 359 (10.6%) under CF-like conditions and 340 (10%) under enriched conditions.

The library compounds were then stratified into two main groups based on their therapeutic/clinical indication, i.e., common antibiotics and drugs that had not previously been reported to have antibacterial activity.

### 2.2. Effect of Growth Conditions on the Anti-P. aeruginosa Activity of Antibiotics

The library included 234 (6.9%) drugs known to possess antibacterial activity (Figure 1). The Pearson correlation coefficient indicated a significant relationship between the findings obtained under CF-like and enriched conditions (r = 0.491, *p* < 0.0001). A total of 99 (42.3%) antibiotics tested were active towards *P. aeruginosa* RP73 under both experimental conditions but to different extents: 43 (18.4%) showed comparable activity under both conditions, and 34 (14.5%) were more active under enriched conditions, whereas 22 (9.4%) showed higher efficacy under CF-like settings (14.5% vs. 9.4%, *p* > 0.05).

A comparable proportion of antibiotics was found to be active exclusively under a growth condition: 54 (23%) were active only under CF-like conditions, and 39 (16.6%) were active only under enriched conditions.

The stratification of data based on the most represented antibiotic classes revealed a positive correlation in the activity observed between experimental settings tested for fluoroquinolones (*p* < 0.0001), tetracyclines (*p* = 0.015), sulphonamides (*p* = 0.0014), beta-lactams (*p* = 0.0004), and antitubercular drugs (*p* < 0.0001) (Figure 2).

The impact of growth conditions on the magnitude of the antibacterial effect of each antibiotic class considered as a whole is summarized in Figure 3A,B. High (70% < x ≤ 90%) or excellent (90% < x ≤ 100%) activity was observed for most of the tetracyclines (cumulative percentage: 72.7% and 90.9%, under CF-like and enriched conditions, respectively) and fluoroquinolones (76.9% and 88.5%, under CF-like and enriched conditions, respectively). Specifically, excellent activity was significantly more prevalent than other magnitudes both under CF-like (63.6% for tetracyclines, *p* at least <0.05; 69.2% for fluoroquinolones, *p* < 0.0001) and enriched (72.7% for tetracyclines, *p* at least <0.05; 88.5% for fluoroquinolones, *p* < 0.0001) conditions. Antitubercular drugs showed a comparable distribution of hits among classes. Sulphonamides showed a higher proportion of hits with excellent activity under CF-like conditions, although not reaching statistical significance (30% vs. 5%, respectively, for CF-like and enriched conditions; *p* > 0.05). Most beta-lactams were inactive regardless of conditions, although a lower percentage was observed under the CF-like setting (42.5% vs. 72.6%, under CF-like and enriched conditions, respectively; *p* < 0.001). Conversely, the activity of macrolides was significantly affected under enriched settings than CF-like settings, as shown by the proportion of inactive antibiotics (7.7% vs. 53.8%, respectively; *p* < 0.05) and those with high or excellent activity (cumulative percentage: 0% vs. 92.3%, respectively, *p* < 0.0001). A trend indicating lower aminoglycoside activity was observed when tested under CF-like conditions, although it did not reach statistical significance (not active drugs: 57.1% vs. 28.6%; excellent activity: 14.3% vs. 28.6%, respectively, in CF-like and enriched conditions; *p* > 0.05).

Considering the impact of growth conditions on the same antibiotic, sulphonamides only showed higher activity under CF-like conditions (50.0% vs. 5.0%, respectively, for CF-like and enriched conditions; *p* < 0.01) (Figure 4A). Conversely, tetracyclines, fluoroquinolones, and macrolides were significantly more effective under enriched settings (45.5% vs. 0%, 26.9% vs. 3.8%, and 38.5% vs. 0%, respectively, for enriched and CF-like settings; *p* < 0.05) (Figure 4A).

When the antibiotics active exclusively under a growth condition were considered, 39 drugs were active only under CF-like conditions: 30 beta-lactams, 4 sulphonamides (Succinylsulfathiazole, Sulfameter, Sulfabenzamide, and Dapsone), 2 aminoglycosides (Amikacin and Ribostamycin), and one each for fluoroquinolones (Pipemidic acid), macrolides (Spiramycin), and antitubercular drugs (Pretomanid), and 24 antibiotics were active only under enriched conditions: 8 beta-lactams (Ampicillin, Ampicillin, Cefamandole, Cefpodoxime proxetil, Ceforanide, Bacampicillin, Ticarcillin, and Meropenem), 6 aminoglycosides (Netilmicin, Bekanamycin, Streptomycin, Gentamicin, Neomycin sulfate, and Paromomycin sulfate), 7 macrolides (Dirithromycin, Midecamycin, Roxithromycin, Acetylspiramycin, Clarithromycin, Erythromycin estolate, and Erythromycin ethyl succinate), and 3 antitubercular drugs (Bedaquiline fumarate, Ethambutol, and Pyrazinamide). A higher proportion of beta-lactams was active only under CF-like settings (41.1% vs. 11.0%, respectively, for CF-like and enriched settings; *p* < 0.0001), whereas an opposite trend was observed for macrolides (53.8% vs. 7.7%, respectively, for enriched and CF-like settings; *p* < 0.05) (Figure 4B).

A comparable proportion of antibiotics showed excellent activity (90% < x ≤ 100%) under both growth conditions (24.3% vs. 23.1%, under CF-like and enriched conditions, respectively), although a different pattern of antibiotics was observed depending on the growth conditions.

Under CF-like conditions, 100% killing was achieved by 37 (15.8%) antibiotics: 13 fluoroquinolones (Lomefloxacin, Delafloxacin meglumine, Prulifloxacin, Pazufloxacin, Cinoxacin, Danofloxacin, Sparfloxacin, Finafloxacin, Trovafloxacin, Gatifloxacin hydrochloride, Norfloxacin, Enoxacin hydrate, and Gatifloxacin), 8 beta-lactams (Mezlocillin, Faropenem daloxate, Cefepime, Cefoperazone, Piperacillin, Doripenem, Aztreonam, and Faropenem), 4 each for tetracyclines (Methacycline, Chlortetracycline, Tetracycline, and Oxytetracycline hydrochloride) and sulphonamides (Sulfamonomethoxine, Sulfamethoxazole, Sulfabenzamide, and Silver sulfadiazine), 2 phenicols (Thiamphenicol and Florfenicol), and one each for antitubercular drugs (Rifapentine), aminoglycosides (Tobramycin), oxazolydinones (Cadazolid), polymyxins (Colistin), beta-lactam/beta-lactamase inhibitors (Sultamicillin), and beta-lactamase inhibitors (Sulbactam). Among these, only 17 (45.9%) were 100% active under enriched conditions also, while 3 (8.1%) were not even active (Mezlocillin, Faropenem, and Sulfabenzamide).

Under enriched conditions, 100% killing was observed for 39 (16.6%) antibiotics: 22 fluoroquinolones, 2 each for beta-lactams, aminoglycosides, phenicols, polymixins, rifamycins, and tetracyclines, and one each for antitubercular drugs, ketolides, macrolides, oxazolidinones, and peptidomimetic antibiotics.

Among these, 17 (43.5%) showed the same activity also under CF-like conditions, while 4 (10.2%) were not even active (Murepavadin, Solithromycin, Netilmicin, and Erythromycin estolate).

### 2.3. Effect of Growth Conditions on the Anti-P. aeruginosa Activity of Compounds with No Known Antibiotic Activity

The Pearson correlation coefficient revealed no significant relationship between findings from CF-like and enriched conditions (r = 0.030, *p* > 0.05) for 3127 compounds belonging to research/therapeutic areas other than “infection” (Figure 1).

A total of 422 (13.4%) drugs were active towards *P. aeruginosa* RP73 under both experimental conditions, although to different extents: 101 (3.2%) compounds showed comparable activity under both conditions, 126 (4%) were more active under CF-like conditions, and 195 (6.2%) were more active under enriched conditions (4.0% vs. 6.2%, *p* > 0.05).

A comparable proportion of drugs was found to be active exclusively under a growth condition: 691 (22.1%) were active only under CF-like conditions, whereas 834 (26.6%) were active only under enriched conditions.

The antibacterial hits found under CF-like conditions (n = 1113) were comparably distributed among the main research areas represented in the compound library (“cancer”: 34.4%; “cardiovascular”: 34.1%; “endocrinology”: 34.8%; “inflammation”: 39.2%; “metabolic disease”: 36.6%; and “neurological disease”: 35.5%) (Figure 5A). However, the proportion of hits with excellent, high, good, moderate, low, and weak activity observed in the “cancer” research area was significantly higher than in other areas (*p* at least <0.05; *p* at least <0.05; *p* < 0.01 except for “neurological disease”; *p* < 0.0001; and *p* at least <0.05) (Figure 5B). Seventeen compounds (1.5%) showed excellent activity (90% < x ≤ 100%): 12 from “cancer” (5-Fluorouracil, Sulforaphane, Tirapazamine, 3-AP, L-SelenoMethionine, Resveratrol, Carmofur, Sulfisoxazole, RRx-001, Ebselen, Panobinostat, and Auranofin), 2 each from “cardiovascular” (Zinc Pyrithione and XL-784), and “neurological disease” (Tolcapone and Radafaxine) areas, and one from “metabolic disease” (Tocofersolan). Specifically, 5-Fluorouracil, Sulforaphane, Tirapazamine, Resveratrol, Carmofur, RRX-01, Ebselen, Auranofin, Zinc Pyrithione, and Tolcapone caused 100% killing.

Under enriched settings, the hit rate was significantly higher in “cancer” compared with “inflammation”, “endocrinology”, and “metabolic disease” (42.7% vs. 34.9%, 33.9%, and 33.8%, respectively; *p* < 0.01) (Figure 6A). The proportion of hits with excellent, high, good, moderate, low, and weak activity observed in the “cancer” research area was significantly higher than in other areas (*p* at least <0.01; *p* at least <0.05; *p* < 0.0001; *p* < 0.0001; and *p* at least <0.01) (Figure 6B). A total of 32 compounds showed excellent activity (90% < x ≤ 100%): 19 from “cancer” (Punicalagin, LTX-315, Tannic acid, Magnolol, Capsaicin, Neticonazole, Bithionol, Curcumin, Incyclinide, Tamoxifen, Brivanib, Econazole, Asciminib, BMS-833923, Gossypol, Ilorasertib, Auranofin, Temoporfin, and 20(*S*)-GinsenosideRg3), 4 each from “cardiovascular” (Zinc Pyrithione, Lovastatin, Eltrombopag, and Eltrombopag-Olamine), and “neurological disease” (Vigabatrin, Vortioxetine, Dasotraline, and Otilonium), 3 from “metabolic disease” (N-Acetyl-L-tyrosine, Tegaserod, and Mitoquinone), and 2 from “inflammation” (Etofenamate and Visomitin). Specifically, LTX-315, Tannic acid, Bithionol, Curcumin, Auranofin, 20(*S*)-GinsenosideRg3, Zinc Pyrithione, Lovastatin, Visomitin, Tegaserod, Mitoquinone mesylate, and Vortioxetine caused 100% killing.

When findings from both settings were comparatively evaluated, the “cancer” research area produced a higher hit rate under enriched conditions than CF-like conditions (42.7% vs. 34.4%, respectively; *p* < 0.001) and a higher proportion of hits with high and good activity (high: 3.1 vs. 1.1, respectively, *p* < 0.01; good: 8.6% vs. 3.6%, respectively, *p* < 0.0001).

CF-like conditions yielded a higher rate of hits with low activity than enriched conditions, both in “metabolic disease” (19.9% vs. 9.6%, respectively, *p* < 0.01) and “neurological disease” (19.7% vs. 13%, respectively, *p* < 0.01).

Under enriched settings, the “neurological disease” research area yielded a higher rate of hits with moderate and good activity than under CF-like settings (moderate: 17.2% vs. 10.9%, respectively, *p* < 0.01; good: 2.7% vs. 0.5%, respectively, *p* < 0.01).

## 3. Discussion

The purpose of this study was to assess how different growth conditions affect an HTS aimed at identifying compounds with therapeutic potential for CF lung infections caused by *P. aeruginosa*. To this end, enriched conditions were compared with those mimicking the nutritional and physical microenvironment of the infected CF lung.

The CF lung is a complex environment where the interaction between nutritional components in CF sputum and opportunistic bacterial infections is balanced. From the first artificial sputum formulation developed by Ghani and Soothill [17], many others have been developed to replicate CF sputum [18]. Considering the variation of formulations and their influence on the investigation of several aspects of bacterial behavior and how these contribute to bacterial fitness and treatment relevance, the first step in choosing a formulation is deciding which is the most comparable to CF sputum.

We are aware that the ASM used here could not be entirely representative of CF sputum because other biologically active components, which could also serve as nutrients, are present in the sputum, including bovine serum albumin, protein-bound iron sources (e.g., ferritin), and bioactive lipids [18]. Nonetheless, we used Sriramulu’s formulation sputum because it allows *P. aeruginosa* to grow in microcolonies, representing an appropriate model of chronic lung colonization useful for evaluating therapeutic procedures [19].

The composition of the library screened in the present study allowed us to assess the impact of different growth conditions on the activity of 234 antimicrobial drugs. Since the library compound mainly consists of non-antibiotic drugs (93.1%), a nutrient-rich medium—i.e., TSB—was used instead of standard cation-adjusted Mueller–Hinton broth.

Our findings showed that growth conditions significantly affect antibiotic efficacy against *P. aeruginosa*. Indeed, although nearly half (42.3%) of antibiotics were active regardless of experimental settings, only a minor proportion (18.4%) showed comparable activity under both conditions. Particularly, tetracyclines, fluoroquinolones, polymyxins, and macrolides were less active under CF-like conditions, while an opposite trend was found for sulphonamides.

In addition, the activity of several antibiotics even manifests under a specific growth condition, with a comparable proportion of antibiotics active only under CF-like (23%) or enriched (16.6%) conditions. However, considering the antibiotic classes, beta-lactams were significantly more effective under a CF-like setting (41.1% vs. 11.0%, respectively, for CF-like and enriched; *p* < 0.0001), whereas an opposite trend was observed for macrolides (53.8% vs. 7.7%, respectively, for enriched and CF-like; *p* < 0.05).

Fluoroquinolones and tetracyclines were found to be the most effective since excellent activity was significantly more prevalent under both CF-like (69.2% and 63.6%, respectively) and enriched (88.5% and 72.7%, respectively) conditions.

Sulphonamides showed a trend of higher activity under CF-like conditions, although this trend was not statistically significant. On the other hand, the activity of macrolides was mostly affected under enriched conditions compared to CF-like settings, with a higher proportion of inactive antibiotics (7.7% vs. 53.8%, respectively; *p* < 0.05) and a higher percentage of antibiotics showing high or excellent activity only under enriched conditions (cumulative percentage: 0% vs. 92.3%, respectively, *p* < 0.0001). Aminoglycosides showed some impact under CF-like conditions, although the differences were not statistically significant, likely due to the limited number of hits tested. Most beta-lactams were inactive regardless of conditions, especially under enriched settings (72.6% vs. 42.5%, under enriched and CF-like settings, respectively; *p* < 0.001).

In trying to better translate the impact that our findings could have on the interpretation of antimicrobial susceptibility testing (AST) results, particular attention must be paid to inhaled (i.e., Tobramycin, Aztreonam, and Colistin), intravenous (i.e., Piperacillin-Tazobactam, Ceftazidime, Ceftazidime-avibactam, Cefepime, and Ceftolozane-tazobactam, Azithromycin, Ciprofloxacin, Levofloxacin, Meropenem, Imipenem/Cilastatin, Doripenem, Aztreonam, and Colistimethate/Colistin), and oral (i.e., Ciprofloxacin and Azithromycin) antibiotics commonly used to improve respiratory symptoms in CF people with *P. aeruginosa* infections.

Antibiotics administered via aerosol directly address the physicochemical conditions observed at the site of infection. Our findings indicated that Tobramycin, Aztreonam, and Colistin are not affected by growth conditions, causing 99.7% or even 100% killing of *P. aeruginosa*. Therefore, in these cases, AST findings could accurately predict the observed activity at the site of infection.

It is worth noting that different trends were observed for antibiotics administered intravenously or orally. Azithromycin, Ciprofloxacin, and Meropenem showed significantly higher activity under enriched conditions, indicating that their activity could be overestimated in classic AST. On the contrary, Ceftazidime, Imipenem, and Doripenem were more active under CF-like conditions, thus suggesting that their effectiveness could be underestimated in AST. Comparable activity was observed for Piperacillin-Tazobactam, Cefepime, Colistin, Aztreonam, and Levofloxacin, causing at least 97.8% killing regardless of the experimental condition.

Overall, our findings confirmed that the experimental conditions simulating those experienced by bacteria during a CF lung infection—i.e., acid pH, low O_2_ tension, and using a synthetic medium resembling CF sputum consisting of type II mucin, DNA, DTPA, NaCl, KCl, egg yolk emulsion, and several amino acids—significantly shape antibiotic activity against *P. aeruginosa*.

Although the present study did not investigate the mechanisms underlying differences in antibiotic activity and the role of ASM composite biomolecules, evidence from previous studies allows us to make some inferences. A hallmark of CF airway is the increased concentrations of mucins—high molecular weight glycoproteins forming a polymeric mesh network—and extracellular DNA (eDNA)—deriving from necrotic and apoptotic leukocytes, release of eDNA networks, or neutrophil extracellular traps—responsible for increased mucus viscoelasticity and decreased mucociliary transport [20]. Consistent with our findings, previous studies have shown that mucus significantly decreases the effectiveness of polymyxins and fluoroquinolones against *P. aeruginosa* [21,22]. Several antibiotic-specific mechanisms could be involved. Mucins offer many potential electrostatic and hydrophobic binding sites for small molecules such as polymyxins—currently the last-resort therapies for infections caused by MDR *P. aeruginosa*—thereby reducing their antibacterial effectiveness [21,23]. Recent reports indicate that eDNA and mucins in the airways can bind to polymyxin lipopeptides (Polymyxin B and Colistin) due to their physicochemical properties, thus reducing their activity [22]. Other suggested mechanisms underlying the fluoroquinolone activity reduction includes mucin modulation of bacterial physiology and mucin reduction in antibiotic uptake by the cells [23].

The effectiveness of several commonly used antibiotics in treating CF infections—e.g., macrolides, fluoroquinolones, and aminoglycosides—may also be affected by biochemical changes in the drug in an acidic environment, such as CF-infected lungs. As the lung disease advances, dehydrated and thickened mucus blocks distal airways, creating microaerobic or frankly anaerobic niches [24]. Lower O_2_ tensions contribute to an acidic pH of the airway surface liquid, where the net positive charge of the antibiotic results in its decreased intracellular penetration and bactericidal activity [8,25].

The ASM formulation includes DTPA, an iron chelator that mimics CF sputum iron availability [26]. Iron depletion could have several, even opposing, effects on antibiotic susceptibility. On the one hand, the synergistic effect between the iron chelator DTPA and the antibiotic significantly reduces *P. aeruginosa* cell viability since iron is essential for all living organisms [27]. On the other hand, low-iron or, more generally, nutrient-limited conditions could impact gene expression [28], revealing new activity for existing antibiotics [29].

In this regard, HTS reveals new anti-*P. aeruginosa* activity for some antibiotics. This is the case for the antitubercular drugs Rifapentine and Rifampicin, which caused nearly 100% activity regardless of the condition tested. The rise in MDR *P. aeruginosa* infections has emphasized the need for new strategies, such as using combinations of antibiotics. For instance, Rifapentine, when combined with Colistin in a dry formulation, improved its activity against both the planktonic cells and biofilm of *P. aeruginosa* [30]. Similarly, Rifampicin combined with Colistin or Fosfomycin was described as a potential adjunctive therapy for Colistin- and carbapenem-resistant *P. aeruginosa*, respectively [31,32]. In addition, Rifampicin potentiates aminoglycoside activity against *P. aeruginosa* strains from CF patients [33]. For the first time in the literature, we found Pretomanid and Macozinone effective under CF-like conditions only, and Isoniazid was more active in CF-like settings than enriched settings, thus revealing their anti-*P. aeruginosa* potential, although to a lesser extent compared with Rifapentine and Rifampicin.

Finally, CF-like conditions impact the transcriptome of *P. aeruginosa*, leading to altered bacterial growth and metabolic activity, finally resulting in enhanced antimicrobial tolerance. Genes showing differential expression—linked to antibiotic efflux or uptake and several metabolic pathways (e.g., TCA cycle, fatty acid catabolism, and amino acid biosynthesis)—can indirectly impact antibiotic efficacy [34].

The compound library we screened consisted mainly of drugs with no documented antibacterial activity (3152 out of 3386 drugs, 93.1%). The HTS assay identified 422 non-antibiotic drugs that were active under both experimental conditions, with similar proportions of compounds showing comparable (CF-like = enriched) or different (CF-like > enriched or enriched > CF-like) activities. Additionally, a comparable proportion of drugs was found to be active only under a specific condition. Interestingly, 691 (22.1%) were active only under CF-like conditions.

The active antibacterial hits under CF-like conditions (n = 1113) belonged to various research areas—i.e., cancer, cardiovascular, endocrinology, inflammation, metabolic disease, and neurological disease—although “cancer” accounted for the highest proportions of hits with excellent and high activity. Most hits with excellent activity (12 out of 17, 70.6%) were anticancer drugs, and 7 of them even caused bacterial eradication.

Several biological properties aimed at damaging the host are shared between tumors and bacterial infections—e.g., high propensity to replicate and spread, resistance to therapy and host immunity, and use of cell–cell communication systems [35,36]—thus explaining why most hits active against *P. aeruginosa* were anticancer drugs. Confirming this, the repositioning of anticancer drugs for infectious diseases [37,38], and vice versa [39], has been reported in the literature. Notably, the anticancer drug 5-Fluorouracil has been proposed for repurposing as quorum-sensing and pyoverdine inhibitors for the antivirulence therapy of *P. aeruginosa* CF infections [40]. The high cytotoxic potential associated with most anticancer drugs represents a reasonable concern for their use as anti-infectives. Anticancer agents might be considered a “last resort” to treat lethal MDR bacterial infections, such as those established in CF lung, where their beneficial effect would outweigh any potential side effect. It is worth noting that 10 drugs with therapeutic indications other than antibiotics—i.e., 5-Fluorouracil, Sulforaphane, Tirapazamine, Resveratrol, Carmofur, RRX-01, Ebselen, Auranofin, Zinc Pyrithione, and Tolcapone—caused the eradication (100% killing) of *P. aeruginosa* under CF-like conditions.

Previous studies have indicated the antibacterial potential of the anticancer drug Tirapazamine and Tolcapone—a catechol-O-methyltransferase inhibitor used in the symptomatic management of idiopathic Parkinson’s disease and hypertension treatment—except against species other than *P. aeruginosa* [41,42,43,44,45]. Notably, Tirapazamine acts as a prodrug activated under low O_2_ concentrations [46], such as those found inside solid tumors—due to poor vascularization—and in CF lungs during chronic infections [24].

Carmofur is an antimetabolite (pyrimidine analog) derivative of 5-fluorouracil used to treat colorectal and breast cancer. In disagreement with our findings, Peyclit et al. [47], using CAMHB, reported carmofur activity against MDR Gram-positive species but not against many Gram-negative species, including *P. aeruginosa*. Thus, this highlights the importance of considering conditions relevant to the site of infection in in vitro assessment of antibiotics’ activity.

RRX-01, a phase III hypoxia-selective epigenetic agent studied as a radio- and chem-sensitizer, triggers apoptosis and overcomes drug resistance in myeloma [48]. RRx-001 exhibits potent anti-tumor activity with minimal toxicity [48]. It is a potent inhibitor of glucose-6-phosphate dehydrogenase and shows potent antimalarial activity [49].

The identified hits may serve as potential “lead compounds” for creating new drugs potentially treating *P. aeruginosa* lung infections in CF patients. In a previous study, we characterized some of those hits for their effectiveness against many *P. aeruginosa* CF strains, cytotoxic potential, and the underlying mechanisms of action [50]. Additional in vivo protection studies are warranted.

## 4. Materials and Methods

### 4.1. Compound Library

The “Drug Repurposing Compound Library” (MedChem Express; Monmouth Junction, NJ, USA) was provided in a 96-well plate format with aliquots of 10 mM stocks of drugs in DMSO or water, stored at −80 °C. It consisted of 3386 bioactive compounds (2342 already approved and 1044 that have reached clinical trial stages in the United States: one drug in Phase I; 606 drugs in Phase II; 372 drugs in Phase III, and 65 drugs in Phase IV) used as drugs with several therapeutic indications, including cancer, neurodegenerative, inflammation, infectious, metabolic, and cardiovascular diseases.

### 4.2. Bacterial Strain and Standardized Inoculum Preparation

The *P. aeruginosa* RP73 strain was used for HTS. Isolated for 16.9 years after the onset of infection in a CF patient from the Hannover cohort [51], this MDR nonmucoid strain is widely used as a prototype for studying chronic illness. HTS was performed using a standardized inoculum. Briefly, some colonies grown overnight on Tryptone Soya Agar (TSA; Oxoid, Milan, Italy) at 37 °C were suspended in sterile saline 0.9% (Fresenius Kabi Italia, Verona, Italy) to reach an optical density at 550 nm (OD_550_) of 0.150. This suspension was diluted 1:10 in sterile saline 0.9% to achieve a final concentration of 1–2 × 10^7^ CFU/mL.

### 4.3. Growth Conditions

HTS assays were carried out under “enriched” and “CF-like” conditions. Enriched conditions consisted of the nutrient-rich general-purpose medium Tryptone Soya Broth (TSB; Oxoid, UK), a pH of 7.2, and an aerobic atmosphere. In CF-like conditions, the physicochemical properties observed in the CF airways [20] were simulated using an ASM at acid conditions (pH 6.8) and under a 5% CO_2_ atmosphere. The ASM composition, according to Sriramulu et al. [19] with some modifications, consisted of (all ingredients were from Sigma-Aldrich unless otherwise specified): 5 g pig stomach type II mucin, 4 g herring sperm DNA, 5.9 mg diethylene triamine pentaacetic acid (DTPA), 5 g NaCl, 2.2 g KCl, 0.1 g Tris-HCl, 5 mL egg yolk emulsion, and 5 g Casamino acids (DifcoTM) per 1 L of water.

### 4.4. Antibacterial HTS Assay

The library was screened at a single concentration point to identify hit compounds against *P. aeruginosa* RP73 [50]. Briefly, 5 µL of the standardized inoculum (0.5–1 × 10^5^ CFU/well) was added to each well of a 96-well polystyrene microtiter plate containing 94 µL of ASM and 1 µL of a 10 mM compound stock solution from the MedChem library, achieving a final drug concentration of 0.1 mM. Uninoculated samples with 1% (*v*/*v*) DMSO (final background in each well) were considered blank. The negative control (100% killing) was prepared using 50% (*v*/*v*) DMSO, whereas the positive control consisted of cells exposed to 1% DMSO. Plates were incubated at 37 °C under 5% CO_2_ for 24 h, and then, the survival rate of planktonic cells was assessed spectrophotometrically using a microplate reader (Tecan Infinite 200 PRO^®^; Tecan Trading AG, Männedorf, Switzerland). The broth culture’s optical density at 550 nm (OD_550_) was measured, and the percentage of surviving cells was calculated compared to the inoculated but untreated control sample (100% survival).

### 4.5. Data Analysis and Interpretative Criteria

The HTS assays were conducted in a single sitting, and the hits obtained were confirmed on a second occasion. A statistical analysis was performed using GraphPad Prism 7.0 software (GraphPad Software, San Diego, CA, USA). The Shapiro–Wilk test assessed the normal data distribution, and differences between the proportions were assessed using Fisher’s exact test. The statistical analysis assumed a confidence level of ≥95%, thus considering *p*-values of <0.05 statistically significant.

The antibacterial activity of library compounds was classified based on the growth reduction compared to the unexposed control sample: (i) not active: <20%; (ii) low: 20% ≤ x ≤ 30%; (ii) moderate: 30% < x ≤ 50%; (iii) good: 50% < x ≤ 70%; (iv) high: 70% < x ≤ 90%; and (v) excellent: 90% < x ≤ 100%. The difference in antibacterial activity of a drug tested under the two growth conditions was considered comparable if −5% ≤ x ≤ 5%.

## 5. Conclusions

The results of the present study suggest that when conducting AST and HTS for drug repurposing, it is essential to consider the physicochemical conditions the pathogen tackles in the human host. This can be achieved by using a testing medium that closely resembles the chemical constitution of human body fluids and the physical conditions observed at the site of infection. By doing so, we can improve the accuracy of and the predictive value of AST and HTS, thus providing a new paradigm for therapeutic intervention for infectious diseases and drug discovery.

In this context, the “CF-like” growth conditions used in the present study not only revealed significant differences in the activity of known antibiotics compared to enriched, AST-like conditions but also identified non-antibiotic drugs that could be repurposed to treat lung infections caused by *P. aeruginosa* in CF patients.

Future studies are warranted to (i) assess potential discordance in response to antibiotics across different *P. aeruginosa* strains; (ii) determine whether CF-like in vitro conditions are predictive of antibiotic efficacy in vivo using animal models; (iii) investigate the mechanisms underlying the impact of growth conditions on antibiotic activity; and (iv) evaluate whether antibiotics selected based on CF-like conditions lead to better clinical outcomes.

## Figures and Tables

**Figure 1 antibiotics-13-00642-f001:**
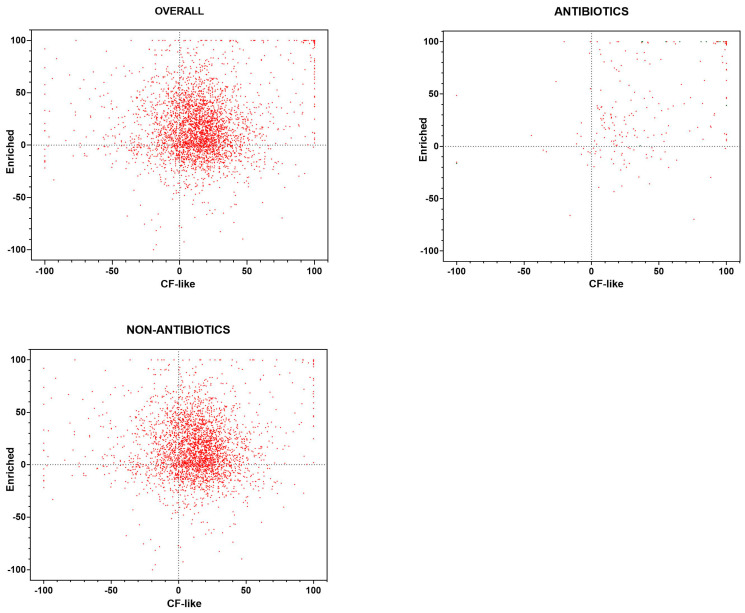
High-throughput screening of the Drug Repurposing Compound Library (MedChem Express). Each compound was tested, under “CF-like” and “enriched” growth conditions, for activity against *P. aeruginosa* RP73 at 0.1 mM, and the survival rate was assessed spectrophotometrically (OD_620_). Results are expressed as the mean percentage of inhibition vs. unexposed control (100% survival). Positive and negative values indicate, respectively, growth reduction and increase. Results are graphed with all compounds included in the library (OVERALL; n = 3386), drugs known to possess antibacterial activity (ANTIBIOTICS; n = 234), and compounds with no known antibiotic activity (NON-ANTIBIOTICS; n = 3127). Twenty-five compounds belong to neither ANTIBIOTICS nor NON-ANTIBIOTICS.

**Figure 2 antibiotics-13-00642-f002:**
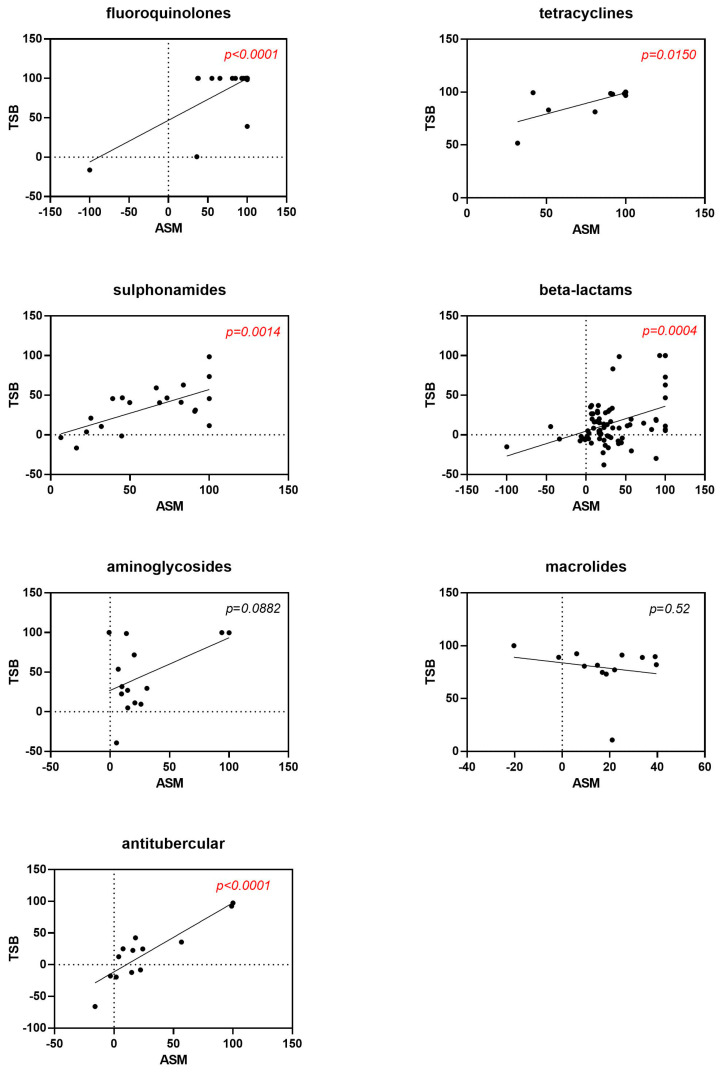
Correlation between antibiotic activity obtained under “CF-like” (ASM) and “enriched” (TSB) growth conditions. Antibiotics are graphed according to the most represented antibiotic classes. Results are expressed as the mean percentage of growth reduction vs. unexposed control (100% survival). Positive and negative values indicate, respectively, growth reduction and increase. Red highlights represent *p*-values < 0.05, indicating statistical significance from Pearson’s correlation coefficient calculations.

**Figure 3 antibiotics-13-00642-f003:**
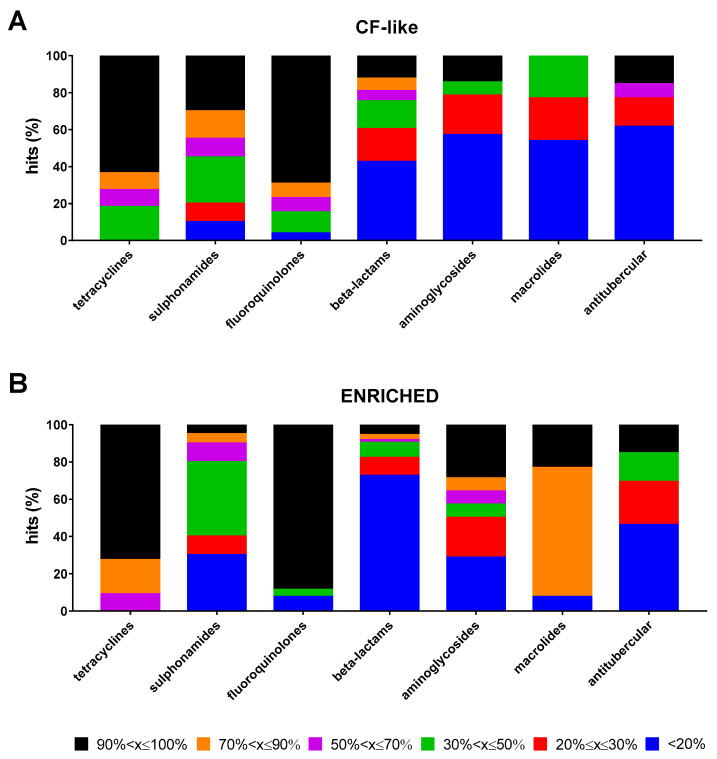
Proportion of antibiotic hits according to classes. The 100% stacked bar graphs show the percentage of hits active towards *P. aeruginosa* RP73 under (**A**) “CF-like” and (**B**) “enriched” growth conditions in each antibiotic class. The magnitude of antibacterial effect (vs. unexposed control) is color-coded as follows: <20% (not active); 20% ≤ x ≤ 30% (low); 30% < x ≤ 50% (moderate); 50% < x ≤ 70% (good); 70% < x ≤ 90% (high); and 90% < x ≤ 100% (excellent).

**Figure 4 antibiotics-13-00642-f004:**
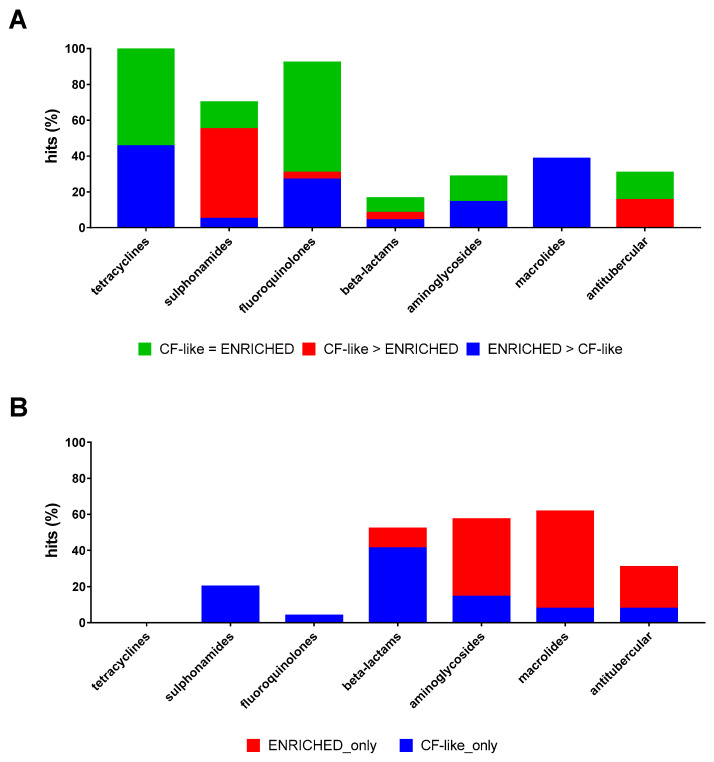
Effect of growth conditions on the proportion of antibiotic hits. The 100% stacked bar graphs show the percentage of hits active towards *P. aeruginosa* RP73 (**A**) both in “CF-like” and “enriched” growth conditions or (**B**) exclusively under “CF-like” or “enriched” growth conditions. The magnitude of the antibacterial effect (vs. unexposed control) is color-coded as follows: <20% (not active); 20% ≤ x ≤ 30% (low); 30% < x ≤ 50% (moderate); 50% < x ≤ 70% (good); 70% < x ≤ 90% (high); and 90% < x ≤ 100% (excellent).

**Figure 5 antibiotics-13-00642-f005:**
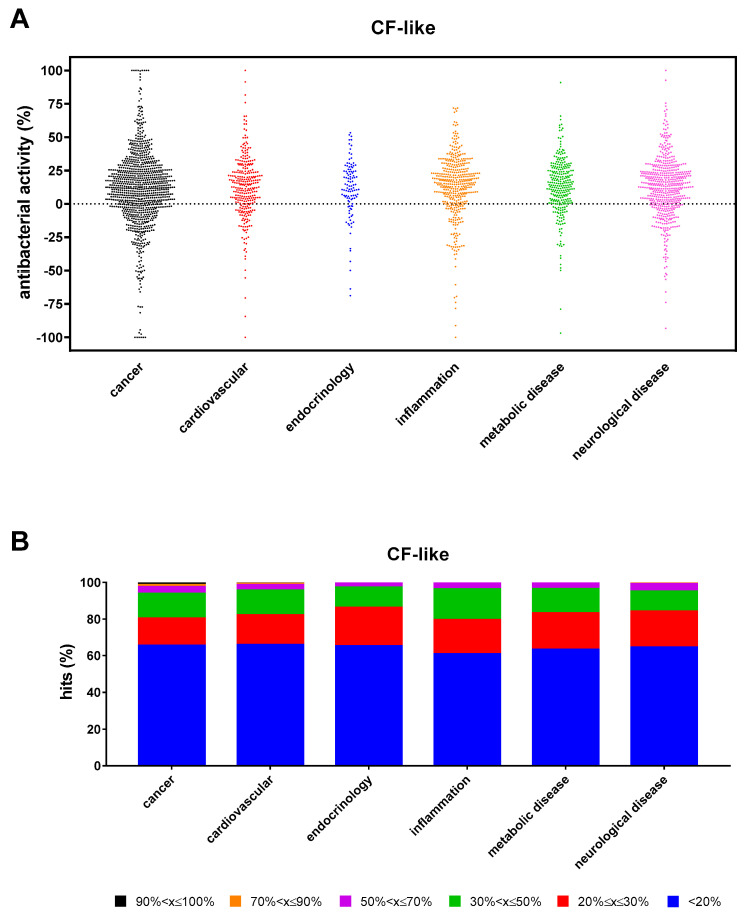
Non-antibacterial hits active towards *P. aeruginosa* RP73 under “CF-like” conditions. (**A**) Proportion of hits with antibacterial activity stratified based on research areas. Positive and negative values indicate, respectively, growth reduction and increase. (**B**) Stacked bar graph showing the percentage of hits active in each research area. The magnitude of the antibacterial effect (vs. unexposed control) is color-coded as follows: <20% (not active); 20% ≤ x ≤ 30% (low); 30% < x ≤ 50% (moderate); 50% < x ≤ 70% (good); 70% < x ≤ 90% (high); and 90% < x ≤ 100% (excellent).

**Figure 6 antibiotics-13-00642-f006:**
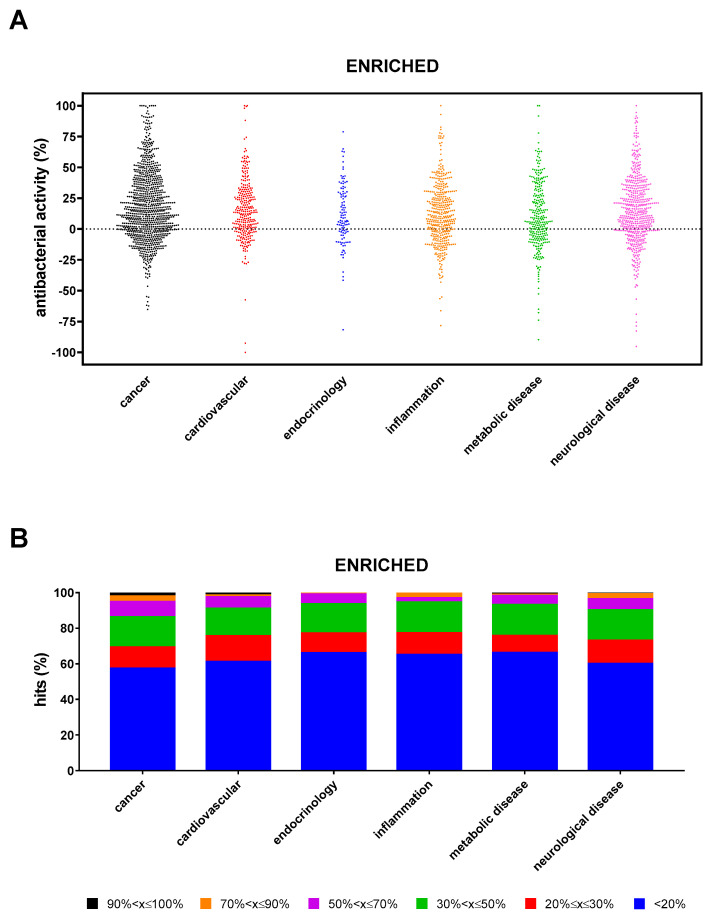
Non-antibacterial hits active towards *P. aeruginosa* RP73 under “enriched” conditions. (**A**) Proportion of hits with antibacterial activity stratified based on research areas. Positive and negative values indicate, respectively, growth reduction and increase. (**B**) Stacked bar graph showing the percentage of hits active in each research area. The magnitude of the antibacterial effect (vs. unexposed control) is color-coded as follows: <20% (not active); 20% ≤ x ≤ 30% (low); 30% < x ≤ 50% (moderate); 50% < x ≤ 70% (good); 70% < x ≤ 90% (high); and 90% < x ≤ 100% (excellent).

## Data Availability

The raw data supporting the conclusions of this article will be made available by the authors on request.

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
