# Peer review of "Impact of Growth Conditions on High-Throughput Identification of Repurposing Drugs for Pseudomonas aeruginosa Cystic Fibrosis Lung Infections"

_antibiotics, 2024, doi:10.3390/antibiotics13070642_

Round 1
Reviewer 1 Report
Comments and Suggestions for Authors
Title: The impact of growth conditions on a high-throughput screening aimed at identifying repurposing drugs with therapeutic potential for cystic fibrosis lung infections caused by Pseudomonas aeruginosa.
Journal: antibiotics
Manuscript ID: 3038741
General opinion
The authors evaluated the effects of growth conditions on the anti-Pseudomonas aeruginosa activity of compounds with different therapeutic indications. They hope to screen out reusable drugs with therapeutic potential for cystic fibrosis pulmonary infection caused by Pseudomonas aeruginosa. However, both experimental design and result analysis are far from achieving this hypothesis. The accuracy of high-throughput screening is high, but the test method is single, the results are poor, and no deeper analysis is carried out. The types of test strains are too few to be convincing. Therefore, I do not recommend the publication of this paper. Comments are as follows:
Major comments
1. The title of this article is too long to clearly illustrate the key points what the authors want to show. It would be good to short and refine the title.
2. The title of this paper is overstated, they only used P. aeruginosa RP73, and could not represent all Pseudomonas aeruginosa.
3. The preliminary research of this article was completely inadequate and the framework of this work is designed not perfect.
4. The purpose of this study was to evaluate the effect of growth conditions on the activity of compounds with different therapeutic indications against Pseudomonas aeruginosa, but the growth conditions included only “CF-like” (artificial CF sputum, pH 6.8, 5% CO2) and enriched (Tryptone Soya Broth, 16 pH 7.2, aerobiosis) conditions, were not sufficient to achieve the purpose.
5. It is suggested that a table should be added to the results to list the selected compounds with anti- P. aeruginosa RP73 aeruginosa activity.
6. In the discussion section, they pointed out that further research needs to confirm its effectiveness in vitro, evaluate its in vivo protective effect and cytotoxicity potential, and explore its potential mechanism of action. It is recommended to add this part of the work in this study.
Minor comments
1. The abstract part does not specify the purpose of the study.
2. The text in the picture is not clear.
3. English proof-reading is surely required throughout
4. It would be good to include additional latest references
5. Please revise the format of the references, such as all the journal names should be abbreviated; the year should be written after the abbreviation of the journal name.
Comments on the Quality of English LanguageExtensive editing of English language required
Author Response
REV 1
(x) I would not like to sign my review report
( ) I would like to sign my review report
Quality of English Language
( ) I am not qualified to assess the quality of English in this paper
( ) English very difficult to understand/incomprehensible
(x) Extensive editing of English language required
( ) Moderate editing of English language required
( ) Minor editing of English language required
( ) English language fine. No issues detected
Yes |
Can be improved |
Must be improved |
Not applicable |
|
Does the introduction provide sufficient background and include all relevant references? |
( ) |
( ) |
(x) |
( ) |
Is the research design appropriate? |
( ) |
( ) |
(x) |
( ) |
Are the methods adequately described? |
( ) |
( ) |
(x) |
( ) |
Are the results clearly presented? |
( ) |
( ) |
(x) |
( ) |
Are the conclusions supported by the results? |
( ) |
( ) |
(x) |
( ) |
Comments and Suggestions for Authors
Title: The impact of growth conditions on a high-throughput screening aimed at identifying repurposing drugs with therapeutic potential for cystic fibrosis lung infections caused by Pseudomonas aeruginosa.
Journal: antibiotics
Manuscript ID: 3038741
General opinion
The authors evaluated the effects of growth conditions on the anti-Pseudomonas aeruginosa activity of compounds with different therapeutic indications. They hope to screen out reusable drugs with therapeutic potential for cystic fibrosis pulmonary infection caused by Pseudomonas aeruginosa. However, both experimental design and result analysis are far from achieving this hypothesis. The accuracy of high-throughput screening is high, but the test method is single, the results are poor, and no deeper analysis is carried out. The types of test strains are too few to be convincing. Therefore, I do not recommend the publication of this paper. Comments are as follows:
Major comments
- The title of this article is too long to clearly illustrate the key points what the authors want to show. It would be good to short and refine the title.
R: The title has been rephrased and shortened.
- The title of this paper is overstated, they only used P. aeruginosaRP73, and could not represent all Pseudomonas aeruginosa.
R: We do not believe that the title is “overstated”. Using a generic “P. aeruginosa” in the title does not mean that ALL P. aeruginosa strains were tested. As always happens in these cases, findings obtained from testing a reference strain COULD BE indicative of a potential activity against other clinical strains of the same species.
- The preliminary research of this article was completely inadequate and the framework of this work is designed not perfect.
R: We disagree with the Reviewer. Even if she/he was right, we found this comment stark, excessively penalizing and not helpful in improving the manuscript. We did not understand what the Reviewer means for “the preliminary research is completely inadequate”. In addition, we believe that it does not exist a work whose framework is “perfect”.
- The purpose of this study was to evaluate the effect of growth conditions on the activity of compounds with different therapeutic indications against Pseudomonas aeruginosa, but the growth conditions included only “CF-like” (artificial CF sputum, pH 6.8, 5% CO2) and enriched (Tryptone Soya Broth, 16 pH 7.2, aerobiosis) conditions, were not sufficient to achieve the purpose.
R: The manuscript is focused on cystic fibrosis. Therefore, we believe that a comparative evaluation between enriched conditions and those relevant to the CF infected lung is more than appropriate to infer adequate and interesting, although preliminary, conclusions.
- It is suggested that a table should be added to the results to list the selected compounds with anti-P. aeruginosa RP73 aeruginosa activity.
R: Three tables (S1, S2, and S3) have been added as Supplemental Material, and cited throughout the manuscript (Lines 98-100).
- In the discussion section, they pointed out that further research needs to confirm its effectiveness in vitro, evaluate its in vivo protective effect and cytotoxicity potential, and explore its potential mechanism of action. It is recommended to add this part of the work in this study.
R: The characterization asked by the Reviewer has already been carried out in a previous study. We referenced this in the “Discussion” section of the revised version (Lines 477-482), adding a new reference [50]. We decided not to change the last sentence of the Conclusions (Lines 547-552) since many potential hits have yet to be studied in depth.
Minor comments
- The abstract part does not specify the purpose of the study.
R: We modified the abstract accordingly.
- The text in the picture is not clear.
R: We need to understand which picture the Reviewer referenced. The font size of the X- and Y-axis titles is sometimes tiny, but they are always readable.
- English proof-reading is surely required throughout.
R: The manuscript underwent proofreading by a native English speaker.
- It would be good to include additional latest references.
R: In the revised version of the manuscript 9 new, updated, references have been added [1,4,11,12,13,14,17,18, 50].
- Please revise the format of the references, such as all the journal names should be abbreviated; the year should be written after the abbreviation of the journal name.
R: The reference style has been modified according to the authors' instructions.
Comments on the Quality of English Language
Extensive editing of English language required
R: The manuscript underwent proofreading by a native English speaker.
Date of this review: 21 May 2024 04:19:06
Reviewer 2 Report
Comments and Suggestions for Authors
Title: Can be more precise as High throughput screening for therapeutic drug for Pseudomonas aeruginosa infection in cystic fibrosis lung.
Abstract: Should be rewrited and add some part of material and method for better understanding, while summarize the results.
Introduction:
1.Review the cause of lung fibrosis and physiological change and the effects of CF to the growth of bacteria
2. Make clear for the purpose that the bacteria (pseudo) cause CF and need antibiotics to treat them or aim to treat bacterial (pseudo) infected CF lung.
3. Review more on potential antibiotics for pseudo
Material and Method:
1.add more references and review on CF-like medium and state some limitations
2.references doses of the antibiotics use in the experiment
3.adding references for AST and HITS assay
Results: Good but the written style is redundant. Could be re write for better understanding.
Conclusion: Appropriate
Comments on the Quality of English LanguageCould be re-write to reduce the redundancy sentences for better understanding
Author Response
REV 2
( ) I would not like to sign my review report
(x) I would like to sign my review report
Quality of English Language
( ) I am not qualified to assess the quality of English in this paper
( ) English very difficult to understand/incomprehensible
( ) Extensive editing of English language required
(x) Moderate editing of English language required
( ) Minor editing of English language required
( ) English language fine. No issues detected
Yes |
Can be improved |
Must be improved |
Not applicable |
|
Does the introduction provide sufficient background and include all relevant references? |
( ) |
( ) |
(x) |
( ) |
Is the research design appropriate? |
( ) |
(x) |
( ) |
( ) |
Are the methods adequately described? |
( ) |
(x) |
( ) |
( ) |
Are the results clearly presented? |
( ) |
(x) |
( ) |
( ) |
Are the conclusions supported by the results? |
( ) |
(x) |
( ) |
( ) |
Comments and Suggestions for Authors
Title: Can be more precise as High throughput screening for therapeutic drug for Pseudomonas aeruginosa infection in cystic fibrosis lung.
R: The title has been rephrased maintaining the focus on the study’s main aim, i.e., evaluating the impact of growth conditions on the anti-P. aeruginosa activity of library compounds.
Abstract: Should be rewrited and add some part of material and method for better understanding, while summarize the results.
R: The aim has been better explicited, and a hint about materials and methods has been added.
Introduction:
1.Review the cause of lung fibrosis and physiological change and the effects of CF to the growth of bacteria
R: The Introduction section has been improved accordingly, adding 5 new references [1, 4, 12, 13, 14].
- Make clear for the purpose that the bacteria (pseudo) cause CF and need antibiotics to treat them or aim to treat bacterial (pseudo) infected CF lung.
The Introduction section has been changed and improved accordingly.
- Review more on potential antibiotics for pseudo
The Introduction section has been changed and improved accordingly.
Material and Method:
1.add more references and review on CF-like medium and state some limitations
R: The manuscript has been changed and improved accordingly (Lines 287-300); two new references [17, 18] have been added.
2.references doses of the antibiotics use in the experiment
R: As stated in the original version, all library compounds – including antibiotics - have been screened at a single concentration point of 0.1 mM. From a technical point of view, it would not have been possible to “personalize” the concentration of selected compounds (e.g., antibiotics).
3.adding references for AST and HITS assay
R: A new reference [50] has been added at Line 512.
Results: Good but the written style is redundant. Could be re write for better understanding.
R: The analytical nature of the findings makes it difficult to rewrite the Results section after proofreading it by a native English speaker.
Conclusion: Appropriate
Comments on the Quality of English Language
Could be re-write to reduce the redundancy sentences for better understanding
R: The manuscript underwent proofreading by a native English speaker.
Date of this review: 27 May 2024 15:30:18
Reviewer 3 Report
Comments and Suggestions for Authors
This paper is very interesting and does address an important question using a well-characterised library of compounds and has reached important conclusions. I feel the results will appeal to medical microbiologists and offer scope for future research. I don't believe any major work is needed to improve the paper and would just detract from the interest. That being said I do have a few minor considerations:
1. Rationale for compound concentration should be outlined.
2. Correlations in figure 2 although positive do look a bit unusual and maybe the overall positive conclusion here is overstated?
3. It would be good to see a few more in-depth growth experiments with some of the most significant compounds to add weight to the findings-maybe some simple growth experiments at a range of concentrations?
Overall, a nice paper.
Author Response
REV 3
(x) I would not like to sign my review report
( ) I would like to sign my review report
Quality of English Language
( ) I am not qualified to assess the quality of English in this paper
( ) English very difficult to understand/incomprehensible
( ) Extensive editing of English language required
( ) Moderate editing of English language required
( ) Minor editing of English language required
(x) English language fine. No issues detected
Yes |
Can be improved |
Must be improved |
Not applicable |
|
Does the introduction provide sufficient background and include all relevant references? |
( ) |
(x) |
( ) |
( ) |
Is the research design appropriate? |
(x) |
( ) |
( ) |
( ) |
Are the methods adequately described? |
(x) |
( ) |
( ) |
( ) |
Are the results clearly presented? |
( ) |
(x) |
( ) |
( ) |
Are the conclusions supported by the results? |
(x) |
( ) |
( ) |
( ) |
Comments and Suggestions for Authors
This paper is very interesting and does address an important question using a well-characterised library of compounds and has reached important conclusions. I feel the results will appeal to medical microbiologists and offer scope for future research. I don't believe any major work is needed to improve the paper and would just detract from the interest. That being said I do have a few minor considerations:
- Rationale for compound concentration should be outlined.
R: We believe it is not possible to find a rationale. Indeed, most previous HTS studies do not explicitly state a specific rationale for the concentration-tested selection. This is mainly due to the inability to know “a priori” the concentrations at which library compounds could show antibacterial activity. We decided to use a negligible volume of the stock compound solution (i.e., 1 µl) to avoid a dilution of the broth volume contained in the well (e.g., 100 µl) that could affect the potential antibacterial activity. Confirming this, the concentration we used (i.e., 0.1 mM) is the same that was tested in several prior reports that evaluated the in vitro anti-P. aeruginosa effects of non-antibiotics (e.g., doi: 10.3389/fmicb.2020.591426).
- Correlations in Figure 2 although positive do look a bit unusual and maybe the overall positive conclusion here is overstated?
R: The Reviewer has not detailed the comment and, therefore, we do not understand the perplexity about the correlations which we found through an appropriate statistical analysis. We do not believe that conclusions arising from the interpretation of data graphed in Figure 2 could be “overstated”, also considering the very high level of significance found (p at least < 0.01), with the only exception for tetracyclines. The sample size could not be optimal for all antibiotic classes, but Pearson’s correlation coefficient calculation also considers this variable.
- It would be good to see a few more in-depth growth experiments with some of the most significant compounds to add weight to the findings-maybe some simple growth experiments at a range of concentrations?
R: A previous publication characterized some of the most promising hits found in our work (doi: 10.1128/spectrum.00352-23). In the manuscript under evaluation, the aim was to evaluate the impact of growth conditions on a primary HTS.
Overall, a nice paper.
Date of this review: 31 May 2024 17:09:17
Round 2
Reviewer 1 Report
Comments and Suggestions for Authors
The manuscripts lacks experiments to verify the effectiveness of compounds and does not meet publication standards.

The language is not concise enough, and the result is unclear.
